# Improving quality of preclinical academic research through auditing: A feasibility study

**Claudia Kurreck**[1], **Esmeralda Castaños-Vélez**[2], **Dorette Freyer**[1], **Sonja Blumenau**[1], **Ingo Przesdzing**[1,3], **Rene Bernard**[1,2,3], **Ulrich Dirnagl**[1,2,3]*

**1** Department of Experimental Neurology, Charité - Universitätsmedizin Berlin, Berlin, Germany,
**2** NeuroCure Cluster of Excellence, Charité - Universitätsmedizin Berlin, Berlin, Germany, **3** QUEST Center for Transforming Biomedical Research, Berlin Institute of Health, Berlin, Germany

* ulrich.dirnagl@charite.de

**Data Availability Statement:** All relevant data are within the manuscript and its Supporting Information files.

## Abstract

How much can we rely on whether what was reported in a study was actually done? Systematic and independent examination of records, documents and processes through audits are a central element of quality management systems. In the context of current concerns about the robustness and reproducibility of experimental biomedical research audits have been suggested as a remedy a number of times. However, audits are resource intense and time consuming, and due to their very nature may be perceived as inquisition. Consequently, there is very little experience or literature on auditing and assessments in the complex preclinical biomedical research environment. To gain some insight into which audit approaches might best suit biomedical research in academia, in this study we have applied a number of them in a typical academic neuroscience environment consisting of twelve research groups with about 100 researchers, students and technicians, utilizing the full gamut of state-of-the-art methodology. Several types of assessments and internal as well as external audits (including the novel format of a peer audit) were systematically explored by a team of quality management specialists. An experimental design template was developed (and is provided here) that takes into account and mitigates difficulties, risks and systematic errors that may occur during the course of a study. All audits were performed according to a pre-defined workflow developed by us. Outcomes were assessed qualitatively. We asked for feedback from participating employees in every final discussion of an audit and documented this in the audit reports. Based on these reports follow-up audits were improved. We conclude that several realistic options for auditing exist which have the potential to improve preclinical biomedical research in academia, and have listed specific recommendations regarding their benefits and provided practical resources for their implementation (e.g. study design and audit templates, audit workflow).

## Introduction

Almost a decade ago, several large pharmaceutical companies blew the whistle on something that had worried the industry for some while: An exceedingly high rate of failures when trying

**Funding:** The Volkswagen Stiftung (Hannover, Germany) support the development of the PREMIER Quality System. The described audits in the manuscript are a Quality tool in that QM System. The funder had no role in study design, data collection and analysis, decision to publish, or preparation of the manuscript.

**Competing interests:** The authors have declared that no competing interests exist.

to reproduce pivotal results of studies performed in academic laboratories, most of which were published in top tier journals [1, 2]. Since then meta-research has identified substantial deficits in the planning, execution, analysis, and dissemination of results of preclinical biomedical research [3]. It appears that preclinical biomedicine is afflicted by a systemic quality problem [4]. A number of remedies have been suggested, some of which appear to gain traction [5] in increasing transparency and data sharing.

Rooted in the beginnings of modern science as an affair of "gentlemen" [6], todays research enterprise is still based largely on trust among scientists [7]. However, given the hyper- competitiveness of the academic system, how much can we rely on the fact that what was reported was actually done? Or exclude that things were done that were not reported? Any laboratory can claim that processes have been implemented to safeguard research quality, for example methods to minimize bias such as randomization and blinding, or the adherence to Standard Operating Procedures (SOP). However, such claims need to be verifiable to be useful to the community. It needs to be demonstrated that what should be done is being done.

Funders and journals increasingly request compliance with their own [8, 9] or general standards, such as making data publicly available, or complying with reporting guidelines such as ARRIVE [10]. Unfortunately, the available evidence suggests that these measures are of limited success [11–14]. It appears that self-reporting of adherence to research quality standard often fails to reflect actual improvements in research practice.

Structured quality management systems (QMS) [15, 16] and more relevant to biomedicine, Good Clinical Practice (GCP) methodology [17] require auditing to avoid discrepancies between reporting and what was actually done in a study. Audits of academic non-clinical research have been suggested a number of times [18–20]. A recent computer simulation suggests that random auditing of research groups can improve research quality, and is highly cost effective [6]. In addition to safeguarding Good Scientific Practice (GSP), auditing may bring additional benefits such as promoting scientific collaboration, the development of common protocols, or fostering communication, transparency and continuous improvement of the scientific processes.

Besides lack of funding, expertise, and a dearth of suitable systems, the prospect of auditing may be among the chief reasons why academia has so far shunned QM systems altogether. In our own experience with ISO 9001 [21], the audits performed by the certification agency were disappointing: Research specific deficiencies in our quality management remained undetected by the auditors untrained in preclinical research, and targeted advice how to improve research quality was consequently very limited.

The Department of Experimental Neurology with its diverse methodological research structure at one of the largest research hospitals in Europe is in an ideal position to investigate whether the controversial instrument of auditing can be practically and effectively applied to an academic research environment to hold researchers more accountable to quality standard in their daily work. Therefore, we have developed a QM system tailored to research [22], in which different forms of assessments and audits are investigated. Specific criteria were favorable cost-benefit ratio, acceptance by our scientists, but also potential efficiency and applicability for other academic biomedical research institutions.

In the following, we will first review which forms of audits we considered suitable for academic preclinical biomedical research and therefore were tested at our department over a time period of several years. Preliminary results of those tests are then presented, and limitations of our approaches discussed.

## Methods

### Site were audits were performed

The research environment of the Department of Experimental Neurology consists of twelve research groups with about 100 researchers, students and technicians. They study fundamental mechanisms of cerebrovascular physiology and pathophysiology using a variety of state-of-the-art technologies and approaches, including molecular biology, cell biology, biochemistry, ultrahigh field magnetic resonance imaging, histology, multi-photon microscopy, *in vivo* modelling of diseases in rodents, cell cultures and brain organoids, and others.

### Definitions of audits and assessments

In research, an audit is a systematic and independent examination of records, documents and processes of an organization (e.g. university, department, research group) to ensure compliance to pre-specified requirements or criteria. An assessment is a demonstration that specified requirements relating to a product, process, system, person or body are fulfilled as defined in ISO 17000 [23].

### Methodology for conducting audits and assessments

A number of different audits and assessment methods with various prerequisites and objectives are in principle relevant at the academic preclinical research setting [23, 24].

Audits and assessments were divided into three categories: self-assessments, internal and external audits. Since every research environment is different, we selected certain types of audits suitable (Table 1), performed different types of audits and assessments (Table 2), while we declared other types as unfit (Table 3) to our specific research settings.

Over three years we have carried out and evaluate different forms of audits and assessments, to define, which of the audit forms are suitable for basic biomedical research. During the first three years working according to DIN EN ISO 9001:2008, we followed the required external certification and monitoring audits, the internal ring audits and the self-assessment with the FMEA [25]. After the change of the QMS from ISO 9001 to PREMIER, other forms of audits and assessments were tried out, such as error reporting using the LabCIRS [24], internal audits of methods, data, documents and processes. The peer audit was performed as a new form of external audit specially tailored to the non-regulated academic environment. The collection of errors and critical incidents with LabCIRS is an ongoing process. The internal audits have been carried out in line with current topics and the need for improvements and standardization of specific processes in our particular research environment.

### Inclusion and exclusion criteria

Before conducting audits and assessments we defined inclusion and exclusion criteria:

**Inclusion criteria:**

- Willingness to participate

- Availability of relevant expertise in the scientific area to be audited

- SOPs available and in use for at least 3 months before the audit

**Exclusion Criteria:**

- Unwillingness to participate

- Insufficient proficiency in the audited area

**Table 1. Types of audits and assessments performed at the Department of Experimental Neurology.** For a more detailed description of specific assessments and audits, see Table 2.

| Assessments / Audits (Aim) | | Who | What | How | Update cycle |
|---|---|---|---|---|---|
| **ASSESSMENTS** | **Risk Assessment, Assessment of Failures and Errors** (identification of risks and needs) | QM competent personnel; scientist | • specific aspects of a scientific project or study<br>• key and support process items for which a risk exists<br>• introduction of new standards | Risk Assessment: *FMEA**<br><br>Error Management: *LabCIRS***, list of errors | • once a year for pre-identified risks and ad hoc when research environment changes significantly |
| **INTERNAL AUDITS** | **Audits of Methods, Data, Documentation, or Processes** (identification of processes or areas that need to be improved) | QM competent personnel; trained scientist | • evaluation of usefulness and effectiveness of implemented measures<br>• identification of gaps | • checklists<br>• documents (e.g. SOPs)<br>• study plan / protocol | • once to several times a year |
| | **Internal Ring Audits of the Charité** (review of formal requirements) | internal auditor of the Charité | preparation for certification and monitoring audits | • checklists<br>• documents (e.g. SOPs) | • once a year, prior to external audits |
| **EXTERNAL AUDITS** | **Peer Audit** (identification of compliance with own requirements) | qualified scientist | • QM system<br>• review of research projects and processes | • questionnaires<br>• checklists<br>• documents (e.g. SOPs)<br>• study plan / protocol | • once during a project |
| | **Certification- and Monitoring Audits (ISO 9001)** (identification of compliance with formal requirements of the norm) | external auditor | • validation of QM system<br>• research projects and processes | • checklists<br>• ISO-norm | • once a year in a certification cycle |

*FMEA: Failure Mode and Effects Analysis [25]

**LabCIRS: Laboratory Critical Incident Reporting System [24].

## Declaration of ethics

The conduct of the audits did not involve individuals. It dealt with research issues, such as data storage or implementation of methods, so that all data from the audits conducted were anonymous at any time. The assessments and reviews reported in our communication are standard quality management procedures as they are carried out in millions of companies and institutions worldwide, and as such are not subject to IRB approval. As a matter of principle, the Charité IRB does not make any statements about non-competence, i.e. there is no need for IRB approval. Audit participants provided no informed written consent.

Table 1 lists all types of audits and assessments performed at the Department of Experimental Neurology as part of our ISO 9001 certification and later quest to establish a bespoke academic QM system. Self-assessments were carried out in the form of risk and error management, as well as internal audits such as method, data, documentation and process audits. A special form of internal auditing were ring audits of the Charité, which were carried out in preparation for the external audits. External audits according to ISO 9001 were conducted in the form of certification and monitoring audits. As a novel form of external audits, a peer audit was devised and performed by experts from another scientific research team (German Mouse Clinic, GMC, Munich Germany). Peer audits are more informal, less structured and aim to evaluate the actual rather than the formal performance.

**Table 2. Description of the assessments and audits listed in Table 1.**

| Assessments / Audits | | When / How often | Method | Objectives |
|---|---|---|---|---|
| SELF ASSESSMENT | Risk Assessment | during the ISO-certification, three times (once a year) | FMEA: All identified risks were recorded by criteria such as flawed methods or results during project progress, project delay, error, loss of data records, etc. The risk score was determined by multiplying the influencing variables' "probability of the occurrence of the risk", "significance for practice when the risk occurs" and "probability of the discovery of the risk that occurred". The higher this figure is, the greater the significance of the risk and the earlier measures had to be taken to avoid it. | • evaluation of processes that run the risk of not fulfilling quality, safety or legal requirements |
| | Assessment of Failures and Errors | ongoing process for six years | • started with a list of errors placed in every laboratory<br>• after two years, implementation of LabCIRS, an anonymous, free, open-source online tool https://github.com/major-s/labcirs, developed by us for this purpose [24]: Error reports are entered anonymously on a computer. QM competent personnel and researchers regularly analyze all error messages; countermeasures are taken with the aim of avoiding systematic errors in the future. Newly established measures are made known to all in the department. | • identification of errors in the daily work routine |
| INTERNAL AUDITS | Audits of Methods, Data, Documentation, Processes | once to several times a year:<br>Methods: 5x<br>Data: 1x<br>Documentation: 5x<br>Processes: 2x | Methods: Core methods (e.g. middle cerebral artery occlusion, preparation of primary cultures of neurons) have been described by scientists through SOPs or working instructions. A member of another working group, but also expert on the described method, examined and questioned the specific implementation, discussed the results and made suggestions for improvement if necessary. In this way, SOPs were checked for deviations or gaps.<br>Data: The archiving procedures of the primary data, and the adherence and practicability of the related SOP were reviewed.<br>Documentation: Project managers, scientist and QM personnel checked whether the documentation requirements were being fulfilled.<br>Processes: A specially developed, detailed experimental/project-planning tool, available as a template in the electronic laboratory notebook, was tested and validated on two projects. | • evaluation of methods, processes and data to be improved<br>• comparison of methods<br>• verification of usefulness and effectiveness of implemented measures<br>• validation of relevant changes to on-going processes or projects<br>• review of compliance with own requirements |
| | Internal Ring Audits of the Charité | during the ISO-certification, three times (once a year) before an external audit | These regular onsite visits were carried out in teams of two to prepare for the certification and subsequent monitoring audits. Spot checks were carried out to ensure compliance with ISO 9001 and included document management, checklists, forms or the results of key performance indicators. | • evaluation of compliance with the ISO-norm |
| EXTERNAL AUDITS | Peer Audit | once, in three working groups of the department | In this unidirectional peer audit, two research groups with expertise in the same field exchanged methods and protocols, reviewed the corresponding procedures and evidence of consistency. They compared methodologies, checked for adherence to protocol or published methodology, best practice details, and discussed potential problems. Scientists with and without training in Quality Assurance, but sufficient background in the audited methods checked for compliance with the detailed experimental / project-planning procedures of the audited project, as specified in the electronic laboratory notebook via a template. | • plausibility checks<br>• comparison of methods<br>• scientific exchange<br>• review of projects and processes |
| | Certification- and Monitoring Audits (ISO 9001) | three times, during the ISO-certification | A certification body verified that our processes, personnel, and management system were compliant with the ISO 9001 requirements of the quality management system (QMS).<br>Following the certification, two monitoring audits were performed, by examining the compliance with requirements of the norm in random checks. | • check whether the QMS is up and running as specified<br>• certification |

Table 2 further details the methods used to carry out these audits, the audit objectives, and the frequency and the time point when they were performed.

**Template for experimental planning of the audited experiment or study.** The Department of Experimental Neurology has developed an experimental design template that takes into account and mitigates difficulties, risks and systematic errors that may occur during the course of a study. Specific questions in the template are geared to reduce the risk of bias in the experiments and raise awareness for project-specific quality issues. This template was developed, used and tested during the peer audit (see Table 2).

**Template audit plan.** An audit plan must be prepared before each audit. In advance, the subjects to be audited and the corresponding schedule are agreed with all participants (see S1 File). In terms of transparency and reliability, it is important to comply with the audit plan, which is the responsibility of the auditing persons.

**Types of audits and assessments, which are not performed at the Department of Experimental Neurology.** Quality management systems use several types of audits and assessments, often to maintain certification and accreditation. Each organization defines which types are best suited to its needs and establishes corresponding audit program objectives. Most available systems are set out to optimize products, services, customer- and supplier relationships all of which have limited applicability in the preclinical academic context. Therefore, several types of audits, such as customer or supplier audits, were not feasible in our environment because we work in a non-regulated, academic, pre-clinical context. We focused only on those considered applicable to our environment in order to identify and analyze potential gaps (such as in protocols, in the implementation of methods, in the storage of raw data etc.).

Table 3 provides a selection of audit types that might be of interest in biotechnological laboratories for example, but are not useful in the biomedical research we conduct.

**Audit workflow.** All types of audit specified in Table 2 were performed according to a pre-defined workflow developed by us (see Fig 1). Audits always included four main steps: Preparation, Execution, Evaluation & Reporting and Proof of Effectiveness. This quality assurance workflow allows the close support of a specific project as well as an entire organization, thus making the improvement at different levels in preclinical research possible. It should be noted that the performance of i.e. internal audits or proof of effectiveness is not necessarily required by professional auditors or QM personnel, but can also be carried out by scientists who are familiar with the audit topic.

**Table 3. Examples for types of audits and assessments, which were considered, but not performed at the Department of Experimental Neurology.**

| Non-Performed Assessments/Audits | not performed because: | Annotations |
|---|---|---|
| *Self-Assessment* with Checklists (Data, Methods, Processes) | • complex of topics too important for self-checks<br>• is carried out in the internal audit by quality personnel or trained scientists | |
| *Internal Audit* Supplier-, Performance-, Compliance Audit | • a supplier audit (aiming to check and improve the current quality and delivery processes) is not important for an academic research institution<br>• performance audits (check of an entity's operations to determine if specific programs or functions are working as intended to achieve stated goals) and compliance audits (review of an organization's adherence to regulatory guidelines) were covered by the certification and monitoring audits | important audits for industry |
| *External Audit* Accreditation Audit | The Department of Experimental Neurology has no accreditation. | only service laboratories are accredited |

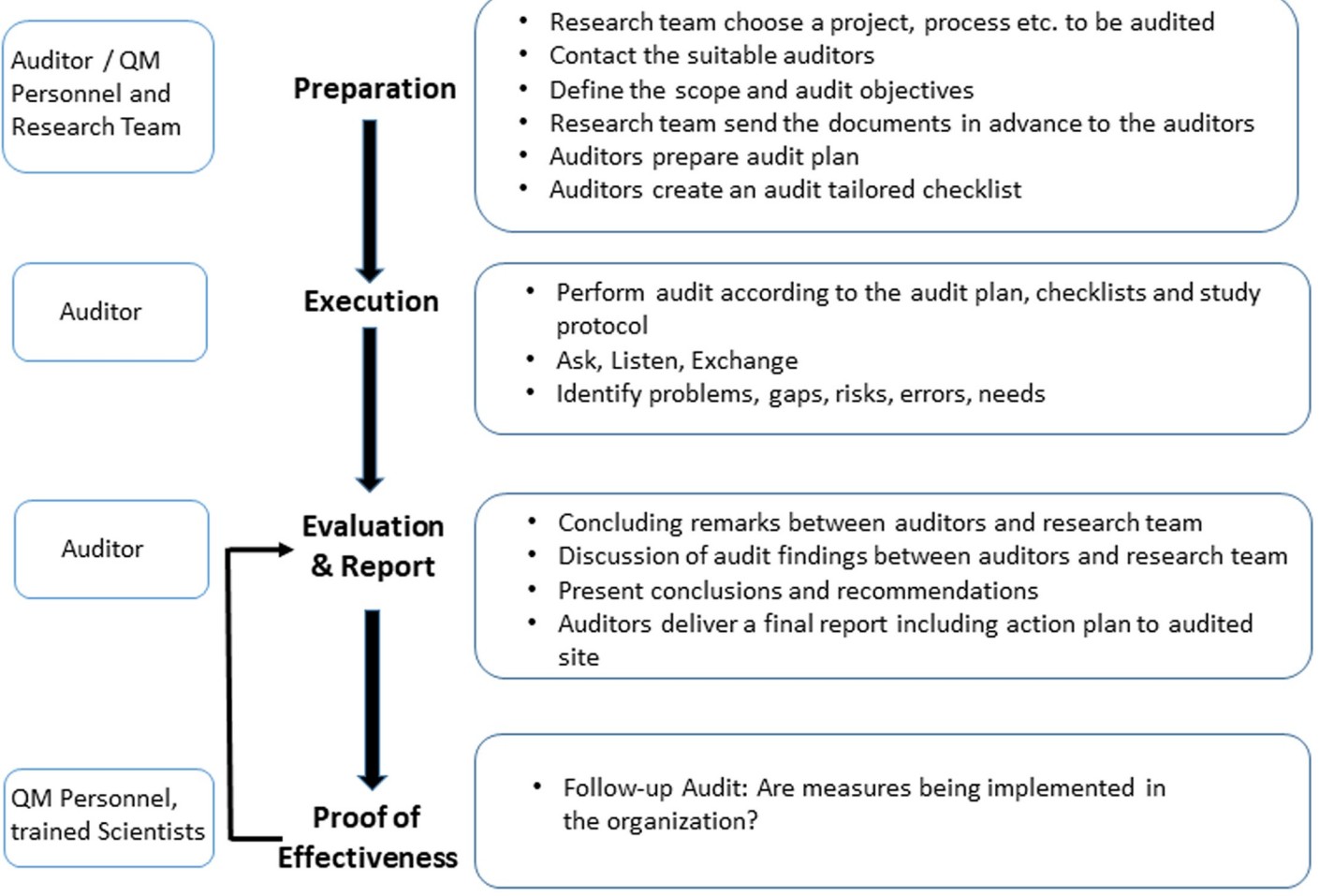

**Fig 1. Audit workflow.** This audit workflow describes the four steps performed during an audit in an academic preclinical research environment, but is universally applicable to all areas in which audits are performed. This audit workflow is a practical example of Deming's PDCA (Plan-Do-Check-Act) cycle, which incorporates the process of continuous improvement.

## Results

As integral part of structured quality management in our department, we carried out and evaluated different types of audits and assessments (such as methods-, data-, process- or peer audits) in an environment typical for preclinical academic research (for details see Tables 1 and 2 above).

Audits were considered successful if they were accepted by the employees, conducted in a clear and non-aversive language, and if the scope and objectives of the audit were clearly defined and if the measures resulting from the audits proved to be effective. Acceptance we defined as agreement of the employees with the objective, focus and manner in which the audit was carried out. The audits were not performed in the style of an examination, but presented as an opportunity for improvement and further development of problems and difficult issues.

The acceptance of the audits was assessed qualitatively. We asked for feedback from participating employees in every final discussion of an audit and documented this in the audit reports. Based on these reports we improved the follow-up audits.

**Table 4. Results of audits and assessments performed in the Department of Experimental Neurology.**

| Assessments / Audits | | Advantages | Disadvantages |
|---|---|---|---|
| SELF ASSESSMENT | Risk Assessment with FMEA | • establishes risk awareness<br>• existing risks identified and recorded<br>• someone must track the measures to prevent risks | • complex and time consuming<br>• risk scores are determined arbitrarily<br>• risks can be avoided or mitigated |
| | Assessment of Failures and Errors with LabCIRS | • establishes awareness for error reporting<br>• allows learning from errors<br>• systematic errors can be prevented<br>• creation of a transparent error culture<br>• anonymous online tool accepted by researches | • needs a person responsible for communicating the reported errors and verifying actions taken to prevent recurrence |
| INTERNAL AUDITS | Audits of Methods, Data, Documentation, Processes | • can be carried out ad hoc by QM competent personnel or trained scientist when needed<br>• identification of usefulness and effectiveness of implemented measures | • at least one responsible person within the organization need to have an overview of where and when internal audits are required |
| | Internal Ring Audits of the institution (Charité) | • prepares for certification and monitoring audits<br>• provides external view<br>• auditors come from the same organization and are familiar with organization-specific processes and institutions | • requires reciprocal audit in another facility of the institution / laboratory<br>• not intended to check in detail any specific procedure<br>• mainly designed for checking the requirements of a certification but not scientific content<br>• needs a person who is trained as an internal auditor |
| EXTERNAL AUDITS | Peer Audit | • external independent expert view<br>• professional exchange at eye level<br>• fosters scientific collaboration<br>• can positively influence the outcome of a project or process<br>• raises awareness among researchers of specific quality issues<br>• provides evidence on the effectiveness and transparency of the scientific process<br>• accepted by researchers | • time consuming for auditors (who have to attend their own research projects) and auditees<br>• travel expenses if the auditors come from another city<br>• not always easy to find suitable audit partners<br>• requires technological and methodological understanding of the research context from the auditor |
| | Certification- and Monitoring Audits (ISO 9001) | • verifies that all requirements of the norm and legal regulations are fulfilled | • the contents of the research processes are only checked to a limited extent<br>• works only as part of a system<br>• high certification costs |

Through our practical experience, we are able to make recommendations for types of audits suitable for preclinical research, which are accepted by researchers, technicians and students. Suitability refers to their importance and relevance in the non-regulated research environment, compliance with regulatory requirement, and the generation of robust results. A summary of the advantages and disadvantages of each type of audit can be found in Table 4. A graphical representation of the number of advantages and disadvantages of audits and assessments see Fig 2.

## Self-assessments

With respect to self-assessment, failure and error management with a structured anonymous error and critical incidence reporting system (LabCIRS [24], freely available at https://github.com/major-s/labcirs) was effective in improving research quality. This was reflected in the effectiveness of the measures resulting from the error reports in LabCIRS, the low rate of recurrence of the same error and increased awareness of specific topics. These included, for

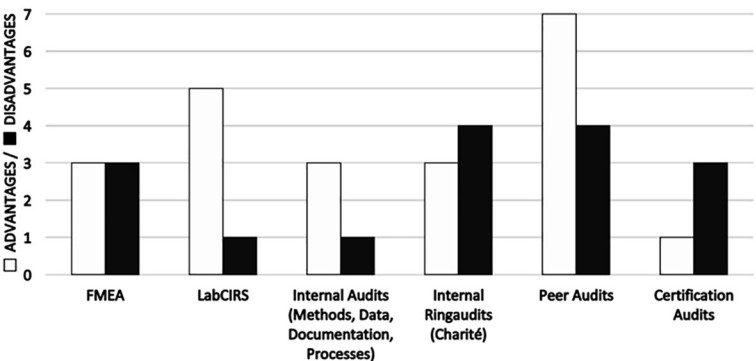

**Fig 2. Graphical representation of the number of advantages and disadvantages of audits and assessments as perceived by members of the department.**

example, improvements in sample identification, comprehensive documentation of experiments in an electronic laboratory book, and the calibration of pipettes. Within three years, a transparent error culture evolved which was built on the overwhelmingly positive experience and acceptance of discussing errors and their prevention. Interest in our LabCIRS has been growing over time, and meanwhile has not been only adopted by other laboratories but is now centrally offered to anyone in the Charité and MDC (LabCIRS BIH) research environment.

We found that the various forms of audits differed greatly with respect to requirement of resources, user experience, efficiency, and fitness for purpose. Interestingly, some of the audit types, such as the Failure Mode and Effects Analysis (FMEA), despite their popularity in other professions and settings (e.g. in the industry or service sectors), revealed severe drawbacks in an academic research setting. For example, many risks in academic research are connected to limitations in continuity in employment contracts and research funds. Finding solutions to this institutional problem via FMEA is very limited. In our experience, FMEA proved to be time consuming, complex, and unpopular among scientists, as the many diverse discussions in the meetings showed.

## Internal audits

With the help of specific and targeted internal audits, risks, errors and gaps between requirements (own, legal, standard) and actual performance were identified, analyzed, evaluated, corrected, monitored and documented. Internal assessments and audits played a key role in the implementation of structured quality assurance measures. Due to their systematic approach to specific issues, they proved to be a powerful mechanism for assessing the validity of methods and data and identifying processes and areas for refinement. With their help, it was possible to initiate a process of continuous improvement and to ensure the effectiveness of implemented measures. An example of such a process was an internal audit for the storage of primary data. Here compliance with the valid Standard Operation Procedure (SOP) and Good Scientific Practice (GSP) was checked and measures to improve the process were implemented. For storage of raw data, the responsible staff (a designated individual in each of 12 working groups) was asked, how, where and in what way the groups store their raw data. When gaps or potential problems were identified, possible solutions were discussed and measures for improvement were formulated. After approximately three months, the implementation and effectiveness of the suggested measures were evaluated. The performance of the audit is

provided as an example in the audit checklist (see S2 File). We therefore recommend internal audits as a useful tool for evaluating critical processes and areas in the research environment.

Internal ring audits of the Charité were useful in preparing for the certification according to ISO 9001.

### External audits

Our department underwent external auditing in two different modes, as part of the ISO 9001 certification and as independent project audit performed by an external peer group. The certification audits by an accredited agency did not have the necessary depth to investigate the operational level of our research processes. As a newly certified research unit, the main focus of the audit was on system implementation and compliance with pre-specified standards. In the follow up discussions, the scientific members of our department perceived these audits as not helpful for their research processes.

A peer audit was performed with the help of experts from the German Mouse Clinic (GMC) in Munich, who conducted an audit of a specific project at our department. Primary objective was to obtain proof of concept of the feasibility of this audit format i.e., a project peer review in a research environment. The audit included external assessment of the project design, plausibility checks, comparison of methods, scientific exchange, and evaluation of the audit format.

Preparing the audit was time-consuming. This format was new for both, the auditors and the audited, and a process had to be developed 'from scratch'. The audit was conducted in a collegial, trusting and open atmosphere. While many audit forms extend their investigations to an entire laboratory or department, peer audits are focused on areas and scientific personnel with direct project involvement.

As opposed to other types of audit during peer audit specifics of the projects were evaluated and discussed in great depth. Using a tailored checklist valuable general and project-specific hints and tips were given. Furthermore, both sides, auditors and participants, exchanged ideas and learned from each other. It must be emphasized, however, that peer audit requires a strong commitment to transparency and academic exchange.

## Discussion

Audits are a central element of quality management systems. Due to the fact that structured quality management is almost unknown in academic preclinical research, there is very little experience or literature on auditing and assessments in this complex research environment. Audits can be carried out by a number of different methods, depending on objectives, scope, criteria, duration, available resources, and context. To gain some insight into which audit approaches might best suit biomedical research in academia, we have applied a number of them, and here we report our experience.

Our experience in an environment typical of biomedical research, gained in a structured way through the introduction of two different QM systems (ISO 9001 [16], and PREMIER [22]) shows that audits can be very useful in pre-clinical research, especially in the evaluation of data, methods, processes and projects. In addition, we found it advantageous for "like-minded people", i.e. colleagues or other researchers who can be trusted in terms of expertise and appropriate and reasonable performance, to conduct audits.

Depending on the size of an individual work group, laboratory, or institution, and the type of research, there are various possibilities for self-assessment, risk assessment and internal and external audits in biomedical basic research. In any case, focus and scope of the audit needs to be predefined in a structured process. To assist the implementation of internal and external

audits, we here provide tools that have been useful in our environment (S1 and S2 Files, Fig 1). In our experience, audits were particularly useful in identifying potential errors and critical issues while projects and studies are planned, conducted, and analyzed. This has been demonstrated, for example, by follow-up audits, which examined the effectiveness of the measures introduced in the previous audit. In this way, a process of continuous improvement can be set in motion and quality in preclinical research improved (see Fig 1).

Peer audits as a special case for external audits are aimed at specific research projects, their methods and approaches. Due to their specificity and direct research focus, peer audits have a higher granularity than certification audits, and the findings and recommendations can be applied directly to specific research processes. We have experienced peer auditing as a bi-directional exchange on an equal footing among like-minded researchers, which ensured an open discussion culture and efficient outcome.

Finding suitable peer auditors may be difficult, since it requires expert knowledge in a specific research area or technique, an open attitude towards quality assessment, and a transparent research culture and exchange. Choice of the peers may bias the outcome of the audit. More importantly, it may be argued that fear of being scooped or loosing intellectual property may prevent scientists from opening up to auditors who are peers and thus potential competitors. First, these fears are unfounded as audits can entirely focus on non-proprietary information, and are covered by non-disclosure agreements. Audits usually revolve around methods and work which was already published by the audited laboratory or group. More importantly, however, is that what appears as a threat is actually an opportunity and a major strength of peer audits. The audited scientists as well as the auditors are united in their quest to improve their own research. During the audit, in particular if performed by a researcher working in the same field with similar methodology, both parties exchange best practice and expose pitfalls and shortcomings. This works in both directions, so that not only the audited laboratory but also the auditor benefit for their own research. In fact, this means that auditors have a selfish interest to engage in something for which a motive, besides research idealism, seems to be missing. In a sense, internal audits can be considered an ideal form of scientific discourse, more powerful than discussions at scientific conventions. Of course, for this to work both parties need to already subscribe to a certain level of openness and transparency. It is our experience that such a spirit is quite common among researchers, and that remaining skepticism as to the benefits and fear of potential threats of audits vanish once the first audit is completed. Our department has benefited from the peer audit conducted. We believe that peer audits have great potential as external reviews, as they can have an immediate positive impact on the outcome of a project or process and can build a trusting and lasting relationship between peer auditing groups.

Based on numerous informal discussions with researchers, students and technicians, regular quality meetings, and exchanges with various stakeholders outside our department, we believe that academic researchers accept audits if they are conducted in a clear and non-averse language and if the scope and objectives of the audit are clearly defined.

While our structured approach to audits has been part of our attempts to implement QM in the academic research environment, it is important to note that audits are not dependent on the existence of a QM system. Audits are a general instrument of quality assurance. Robustness and reproducibility of research results can only be achieved if reporting reflects actual practice accurately. In this context, audits are a simple and valid instrument to check the consistency between scientific practice and reporting. Quality assurance in preclinical research is based on measurement and continuous improvement to evaluate the effectiveness of the applied quality measures. To maintain a high level of performance in research work, it is necessary to react to changes in internal and external requirements and to create new solutions. Audits are one possibility to achieve these objectives.

Given the lack of structured QM in pre-clinical research, we suggest that audits should be adapted for use in this area without necessarily relying on an existing QM system. The systematic approach to QM in academic preclinical research that we are currently developing [22] includes an audit module that can be used independently and provides practical advice on how to conduct audits and assessments.

## Limitations

Our study has a number of weaknesses and limitations. Although we believe that our research environment is typical for a large fraction of biomedical research laboratories in academia, we can only present anecdotal evidence from one department, one area of research (neuroscience), and one country (Germany). This precludes formal statistical analysis, and unfortunately does not provide insights from other types of laboratories, such as those in the physical or chemical sciences. Our approach was also not preplanned and preregistered, and our outcome assessment is highly subjective. In addition, the number of audits performed was limited. Audits were prepared and conducted by a team of QM professionals in a research environment preconditioned to issues of QM. We nevertheless believe that our experience can be helpful to many stakeholders (researchers, institutions, funders) as to our knowledge it is the first attempt to formally approach this issue. Our results can inform larger and more systematic studies in the future and stimulate discussion.

## Conclusions

Despite these limitations, from our hands-on experience with various approaches to auditing in preclinical academic research, and the feedback received from multiple stakeholders, we conclude the following:

- Audits, as well as any kind of assessment can only function properly if those audited understand the concept and its merits and participate voluntarily.

- Structured error management with a laboratory critical incidence system is an effective self-assessment activity, which can be implemented relatively easily. Beyond helping in the recurrence of reported errors, it is a powerful tool to introduce a non-punitive error culture.

- Internal process-, document- and data audits are a straightforward approach to identify, analyze, and correct risks, errors and gaps in projects. They safeguard the validity of methods and data and help to identify areas that need improvement. These internal audits foster compliance with the complex legal and regulatory frameworks of biomedical research (animal protection laws, 3R, safety regulations, reporting guidelines like ARRIVE, etc.).

- Depending on the size of the organization, internal audits can be done several times a year and in particular in critical stages of a project. Internal audits are also useful at the organizational level for reviewing quality assurance processes.

- Peer audits are a promising novel tool to solicit external feedback and foster professional exchange of ongoing projects at eye-level. They are very effective in fostering the improvement of methods or processes.

- We recommend conducting at least one peer audit during a project's lifetime, especially if specific methodological challenges need to be solved and an exchange with colleagues is desired.

- Auditing contributes to transparency. In general, and peer audits in particular, auditing might serve as fundamental processes of open and transparent science in the future.

- Careful and collegial communication of expectations and goals is a prerequisite for successful audits.

Audits are an opportunity to safeguard research quality and transparency, not instruments of control. Audits can increase trust between researchers, and between researchers and the public, in particular at times where this trust has eroded. Specific forms of auditing may be selected or further developed to benefit all stakeholders in the system, researchers, institutions, funders, scholarly societies, journals, and the public. We conclude that several realistic options for auditing exist which have the potential to improve research, and have listed specific recommendations regarding their benefits and implementation. Further systematic studies are needed to select and adapt auditing approaches acceptable to and practical for all stakeholders in the academic research enterprise and evaluate how their implementation affects research quality.

## Supporting information

**S1 File. Template audit plan.**
(PDF)

**S2 File. Checklist internal and external audit.**
(PDF)

**S3 File. Table: Baseline characteristics of participants.**
(DOCX)

## Acknowledgments

We thank Valerie Gailus-Durner, Claudia Stöger, Adrian Sanz Moreno, Julia Calzada-Wack (German Mouse Clinic, GMC, Munich, Germany) for performing the peer audit and stimulating discussions.

Nikolas Offenhauser (VIOMEDO, Berlin, Germany) helped us with the ISO certification process.

We gratefully acknowledge the support of the Volkswagen Stiftung (Hannover, Germany) for support in developing the PREMIER Quality System.

## Author Contributions

**Conceptualization:** Claudia Kurreck, Ulrich Dirnagl.

**Data curation:** Claudia Kurreck, Ulrich Dirnagl.

**Formal analysis:** Claudia Kurreck, Ulrich Dirnagl.

**Funding acquisition:** Claudia Kurreck, Ulrich Dirnagl.

**Investigation:** Claudia Kurreck, Ulrich Dirnagl.

**Methodology:** Claudia Kurreck, Dorette Freyer, Sonja Blumenau, Ingo Przesdzing, Ulrich Dirnagl.

**Project administration:** Claudia Kurreck, Ulrich Dirnagl.

**Resources:** Claudia Kurreck, Ulrich Dirnagl.

**Software:** Claudia Kurreck, Ingo Przesdzing, Ulrich Dirnagl.

**Supervision:** Claudia Kurreck, Ulrich Dirnagl.

**Validation:** Claudia Kurreck, Ulrich Dirnagl.

**Visualization:** Claudia Kurreck, Ulrich Dirnagl.

**Writing – original draft:** Claudia Kurreck, Ulrich Dirnagl.

**Writing – review & editing:** Claudia Kurreck, Esmeralda Castaños-Vélez, Rene Bernard, Ulrich Dirnagl.

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
