## [Decision Letter · Decision Letter 0]

7 Jul 2020

PONE-D-20-06568

Improving quality of preclinical academic research through auditing: A feasibility study

PLOS ONE

Dear Dr. Dirnagl,

Thank you for submitting your manuscript to PLOS ONE. After careful consideration, we feel that it has merit but does not fully meet PLOS ONE’s publication criteria as it currently stands. Therefore, we invite you to submit a revised version of the manuscript that addresses the points raised during the review process.

The authors are advised to address the concerns raised by reviewers.

We look forward to receiving your revised manuscript.

Kind regards,

Ghulam Md Ashraf, Ph.D.

Academic Editor

PLOS ONE

Journal Requirements:

2. In the ethics statement in the manuscript and in the online submission form, please provide additional information about the records used in your study.

Specifically, please ensure that you have discussed whether all data from the audits conducted (by you or other researchers) were fully anonymized before you accessed them and/or whether the IRB or ethics committee waived the requirement for informed consent.

If audit participants provided informed written consent to have these data used in research, please include this information.

3. We note that you have included your manuscript title page as a separate document.

Please ensure that you include a title page within your main document.

You should list all authors and all affiliations as per our author instructions and clearly indicate the corresponding author.

Reviewers' comments:

Reviewer's Responses to Questions

**Comments to the Author**

1. Is the manuscript technically sound, and do the data support the conclusions?

Reviewer #1: Partly

Reviewer #2: Yes

2. Has the statistical analysis been performed appropriately and rigorously? 

Reviewer #1: No

Reviewer #2: N/A

3. Have the authors made all data underlying the findings in their manuscript fully available?

Reviewer #1: Yes

Reviewer #2: Yes

4. Is the manuscript presented in an intelligible fashion and written in standard English?

Reviewer #1: Yes

Reviewer #2: Yes

5. Review Comments to the Author

Reviewer #1: 1. Authors mentioned that they had analyzed the potential gaps like storage of raw data. Authors must mention how they had analyzed it and how they had addressed it. Because raw data is the most crucial element when we talk about research reproducibility.

2. Statistical analysis is missing in the manuscript as how this new audit version is better and how it has addressed the potential gaps. Accuracy, preciseness etc are missing elements.

3. References should be in the same parenthesis. For instance, it will be as per format and appropriate to write [1,2] rather than [1] [2].

4. This sentence needs rearrangement to become more clear “…….how much can we rely on that what was reported was actually done?”

5. The sentence is repeated in manuscript “audits are resource intense and time consuming, and due to their very nature may be perceived as inquisition”

6. Subheading of “The Department of Experimental Neurology” is inappropriate under Methods section. It should be replaced with suitable alternative.

7. Terminologies like in vitro, in vivo should be in italics.

8. In section ‘Methodology for conducting audits and assessments’, authors have directly jumped from Table 1 to Table 3.

9. Punctuation marks should be appropriately used throughout the manuscript.

10. Authors have mentioned “Over the years”, please mentioned precisely the number of years that the team had devoted in this study.

11. Additional bullet point in exclusion criteria.

12. Wrong file uploaded in “S2_File_Checklist_Internal_and_External_Audit.pdf”

13. PDF on the last page as Preprint is also missing.

14. As the manuscript is quite inclined towards re-validation and expression of significance of LabCIRS BIH and PREMIER quality management system. It should be available for the public domain for the assessment, not on the intranet. Or in case if it is not possible to make it publically available, then also there should be proper statement that its access can be provided to any official authority if required for assessment.

15. While performing the audit of the research work, the major reason for the academicians for back-footing is the trust issue on the auditor. How this trust issue can be resolved in this case? Because if it is always a very known auditor, then audit can definitely be biased to some extent, and if the auditor is fully unknown, then the threat of IPR will be there.

16. References needs to be thoroughly rechecked to eliminate any such errors like “PMCPMC5896971”

Reviewer #2: The manuscript "Improving quality of preclinical academic research through auditing: A feasibility study" is a well conceived and thoughtful piece of research done by the authors highlighting some of the crucial issues to conduct biomedical research in an unbiased and transparent manner. The authors have reasonably explained all the pros and cons of the auditing process in a biomedical research setup. However I have a few minor concerns regarding the work presented here;

1. The authors don't mention anything about any possible preliminary steps that can be taken to audit research work in physical and chemical sciences.

2. Provide reference for ISO 17000 in "Definitions of Audits and Assessments"

3. What is the 3rd point in Exclusion Criteria in the "Inclusion and Exclusion Criteria" section.

4. The auditing processes presented here is multi-step and pretty complicated so persons doing the audit process needs to be trained and well versed with the whole audit process. I would therefore recommend to include a short write up on the necessary training of personnel prior to the auditing.

5. Some of the data in "Results of audits and assessments performed in the Department of Experimental Neurology in Table 4" can be presented in a statistical graph format to improve the readability and comprehensibility for a wider audience.

6. PLOS authors have the option to publish the peer review history of their article (what does this mean?). If published, this will include your full peer review and any attached files.

Reviewer #1: **Yes: **Dr. Rajeev K. Singla

Reviewer #2: No

---

## [Author Response · Author response to Decision Letter 0]

7 Sep 2020

Answer to the reviewers 

Dear Editors, 

The authors of the paper would like to thank the reviewers for their valuable comments and helpful suggestions. We hope that the modifications we have made to the manuscript based on them will make it more readable and understandable. We have tried to address all the issues raised, and will respond to them and highlight modifications to the manuscript point by point below. 

If you have any further queries, please do not hesitate to contact us at any time.

Sincerely

Prof. Dr. Ulrich Dirnagl

In behalf of the authors.

You will find all our answers to your questions in the rebuttal letter.

---

## [Decision Letter · Decision Letter 1]

2 Oct 2020

Improving quality of preclinical academic research through auditing: A feasibility study

PONE-D-20-06568R1

Dear Dr. Dirnagl,

We’re pleased to inform you that your manuscript has been judged scientifically suitable for publication and will be formally accepted for publication once it meets all outstanding technical requirements.

Kind regards,

Ghulam Md Ashraf, Ph.D.

Academic Editor

PLOS ONE

Additional Editor Comments (optional):

Reviewers' comments:

Reviewer's Responses to Questions

**Comments to the Author**

1. If the authors have adequately addressed your comments raised in a previous round of review and you feel that this manuscript is now acceptable for publication, you may indicate that here to bypass the “Comments to the Author” section, enter your conflict of interest statement in the “Confidential to Editor” section, and submit your "Accept" recommendation.

Reviewer #1: All comments have been addressed

Reviewer #2: All comments have been addressed

2. Is the manuscript technically sound, and do the data support the conclusions?

Reviewer #1: Yes

Reviewer #2: Yes

3. Has the statistical analysis been performed appropriately and rigorously? 

Reviewer #1: N/A

Reviewer #2: Yes

4. Have the authors made all data underlying the findings in their manuscript fully available?

Reviewer #1: Yes

Reviewer #2: Yes

5. Is the manuscript presented in an intelligible fashion and written in standard English?

Reviewer #1: Yes

Reviewer #2: Yes

6. Review Comments to the Author

Reviewer #1: The authors have carefully addressed both the reviewer's observations and meticulously revised the manuscript with proper justifications at relevant places.

Reviewer #2: (No Response)

7. PLOS authors have the option to publish the peer review history of their article (what does this mean?). If published, this will include your full peer review and any attached files.

Reviewer #1: **Yes: **Dr. Rajeev K. Singla

Reviewer #2: **Yes: **Dr. Pinaki Misra, Ph.D

---

## [Editor Report · Acceptance letter]

6 Oct 2020

PONE-D-20-06568R1 

Improving quality of preclinical academic research through auditing: A feasibility study 

Dear Dr. Dirnagl:

I'm pleased to inform you that your manuscript has been deemed suitable for publication in PLOS ONE. Congratulations! Your manuscript is now with our production department. 

Kind regards, 

on behalf of

Dr. Ghulam Md Ashraf 

Academic Editor

PLOS ONE